
# Calorimetry for active systems

Pritha Dolai[1*], Christian Maes[1] and Karel Netočný[2]

**1** Instituut voor Theoretische Fysica, KU Leuven, Belgium
**2** Institute of Physics, Czech Academy of Sciences, Prague, Czech Republic

⋆ pritha.dolai@kuleuven.be

## Abstract

We provide the theoretical basis of calorimetry for a class of active particles subject to thermal noise. Simulating AC-calorimetry, we numerically evaluate the heat capacity of run-and-tumble particles in double-well and in periodic potentials, and of systems with a flashing potential. Low-temperature Schottky-like peaks show the role of activity and indicate shape transitions, while regimes of negative heat capacity appear at higher propulsion speeds. From there, a significant increase in heat capacities of active systems may be inferred at low temperatures, as well as the possibility of diagnostic tools for the activity of self-motile artificial or biomimetic systems based on heat capacity measurements.

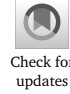
# 1 Introduction

The influence of heat on chemical reactions was already breaking ground in the 18th century [1]. Yet and despite growing successes of molecular theory, physics understanding remained incomplete for the specific heat of gases. That played a crucial role in the emergence of quantum mechanics and its applications to condensed matter physics. Since then, heat capacity is often depicted as a material constant, changing only by modifying volume or by varying temperature or other intensive parameters. It is however natural that activity may play a role as well, as equipartition is easily violated by driving or active forces [2]. For example, we expect that a tissue changes its heat capacity when it is acted upon by molecular motors, or a transmission device when undergoing random potential changes while maintaining relatively large heat fluxes or electric currents. Similarly and not thoroughly explored so far, we believe that the heat capacity of living material will be different depending on its activity (as seen in metabolic rates or biological functioning), [3, 4]. The results of the present paper make the question precise and present the first systematic evidence, as it starts the computation and study of heat capacities for active matter.

Quantitative explorations of heat due to biological functioning or as a function of metabolic rates and changes therein are studied under the heading of bioenergetics [5]. References on measuring heat production in bacterial reservoirs include [6–8]. In general, however, not much got systematized on the theory side of condensed matter and nonequilibrium statistical mechanics. The same holds true for active meta- and morphing materials where thermal properties and functionalities may depend on nonequilibrium driving [9, 10]. It is again important there to quantify the relation between heat and temperature, including now at very low temperatures.

The present paper takes this question of defining and computing heat capacities to the paradigmatic case of active systems [11]. Active matter [12–15] is of growing interest for new materials and functionalities, and it appears important to scan a large temperature range for its thermal properties. Thermal properties of active particles have of course been widely discussed, including [6, 8, 16–22]. Yet, heat capacities, the traditional window on "active" degrees of freedom, have not been calculated, let alone explored there. We suspect that the lack of experimental work on heat properties of active systems is mainly due to the absence so far of a theoretical framework and model calculations, a gap the present paper wants to fill. In the same way, while Life processes happen on a much smaller window of temperatures, we wish to understand how the heat capacities of bio-materials depend on activity parameters. Particle models obviously only shadow the complex mechanisms of Life or of active materials that cannot be sustained in thermodynamic equilibrium but, by combining exact results and simulation, they are capable of highlighting important phenomena.

The used prototypical models feature flashing potentials and run-and-tumble particles (RTPs) subject to periodic and confining potentials. RTPs go under the heading of self-propelled particles, whether biological or artificial, on the colloidal (mesoscopic) level of description. In all cases, except for two exact calculations illustrated in Appendix A, we obtain

the heat capacity by combining simulation and numerical work, applying for the first time the scheme of AC-calorimetry [23–25] to active systems. We investigate the role of propulsion speed and of tumbling or flashing rates on particles that are either confined or move on a periodic landscape. The main results are visualized in plots of the heat capacity. Each time, that is followed by discussion and explanations. Specific issues concern the relation with equilibrium systems and the occurrence of negative heat capacity. In the end, we also consider the heat-related entropy for these systems.

While the models are for simplicity restricted to one dimension, we do not believe that is a serious restriction as we are not probing thermal properties near phase transitions. In fact, we are including a study of active particles in a double-well potential that imitates, in the usual mean-field sense, higher-dimensional active particle models.

For simplicity we consider here translational motion only, ignoring e.g. activity-induced vibration or rotation. The dynamical variable is a scalar, like the position on the real line without considering inertial degrees of freedom, and the irreversible work done on the particle is by the active forces or by flashing the potential.

In the next section, we recall elements of nonequilibrium calorimetry as we would use in realistic AC-calorimetric measurements but applied to run-and-tumble particles. The following sections are devoted to the results for the heat capacity of run-and-tumble particles, which sometimes reduce to motion in a flashing potential. The final section draws some general conclusions. In the Appendix A, the computation for an exactly solvable case is presented.

## 2 Heat capacity for an active gas

By an active gas, we mean a collection of point-like quasi-independent components moving in a viscous and thermal equilibrium environment via mechanisms of self-propulsion. There exist various models that are thought relevant for bacterial motion (E. Coli in particular, see e.g. [26–28]), for artificial self-motile colloidal particles [29–31], and more generally, as theoretical abstractions of functional entities working within active or morphing matter [32, 33]. For the purpose of the present paper, which is opening the field of active calorimetry, we have chosen to work with a simple model dynamics, for so-called run-and-tumble particles. From the questions and the used methods, it is clear that the work has a much wider scope and applicability.

### 2.1 Model

We are considering a standard model of effectively independent active particles, known under the name of run-and-tumble particles (RTPs). In one dimension, they keep a nonzero propulsion speed $v$, and tumble (change direction) at random moments. The overdamped dynamics for the position $x_t$ at time $t$ is

$$\gamma \dot{x}_t = v \sigma_t - U'(x_t) + \sqrt{2\gamma T}\, \xi_t \tag{1}$$

under standard white noise $\xi_t$. There is a dichotomous noise $\sigma_t$ as well, not depending on temperature $T$ nor on the location of the particle: the $\sigma_t \in \{+1, -1\}$ is flipping at fixed rate $\alpha > 0$. The prime in $U'$ denotes a derivative with respect to $x$. In general, we use $E_0$ to denote its magnitude. Using $\alpha^{-1}$, $v/\gamma\alpha$ and $E_0$ as the unit of time, length and energy, respectively, the (1) can be reduced to a dimensionless form,

$$\frac{dX_\tau}{d\tau} = \tilde{\sigma}_\tau - \frac{E_0 \gamma \alpha}{v^2} \Psi'(X_\tau) + \sqrt{\frac{2\gamma T \alpha}{v^2}}\, \xi_\tau \tag{2}$$

where $\tau$, $X_\tau$ and $\Psi$ are the dimensionless time, position and potential respectively. The dichotomous noise $\tilde{\sigma}_\tau$ now flips at constant rate one. From (2), we can understand what relations matter for the dependence of e.g. thermal features. Of course, the shape of the potential $\Psi$ matters a great deal, and comparing with the equilibrium case where $v = 0$ is not possible that way. In many applications at fixed $E_0$, it is also more insightful to study the dependence on $v$ and $\alpha$ explicitly. We will therefore not insist on (2) and we follow the writing (1), where, to set the time-scale, we put friction $\gamma = 1$ (and we already put Boltzmann's constant $k_B = 1$).

The run-and-tumble dynamics (1) is presented here and elsewhere as paradigmatic for active particles; see [12]. Its origin is in discussions of dichotomous noise, and most often for greater simplicity, the dynamics appears without thermal noise ($T = 0$). Finite thermal noise may be relevant, especially for artificial self-propelled transport on nanoscales, and that is also why the low-$T$ behavior is focus of our attention. More generally, it is quite natural, in studying thermal properties of active components, to introduce a heat bath at a finite temperature. After all, active systems are open, and active components exchange energy with and dissipate into a thermal environment. Whether the contribution to that heat from (changes in) translational motion is measurable, is not so clear however, especially for bacteria. For artificial systems, like in [29,31], the situation and role of thermal noise is more clear.

## 2.2 Heat and work

All calorimetry, be it in or out of equilibrium, starts with the First Law of Thermodynamics. Since we have an energy and a heat bath at temperature $T$ in (1), we can consider the heat flowing to it.

For the instantaneous expected heat flux, we need to take the change of energy minus the work done on the particle. For the energy change $U(x_{t+dt}) - U(x_t)$, we apply Itô's Lemma and get it to equal

$$U(x_t + dx_t) - U(x_t) = \left[ \left( v\sigma - U'(x_t) \right) U'(\eta, x_t) + T U''(x_t) \right] dt + \sqrt{2T} \, U'(x_t) \xi_t dt \quad (3)$$

(with, more mathematically correct, $\xi_t dt = dW_t$ for the standard Wiener process $W_t$). The expected work done per unit time on the particle for locomotion is $\dot{w}(x; \sigma) = \sigma v(-U'(x) + \sigma v)$. The expected energy change conditioned on $x_t = x$ is just given by (3) but without the noise term. Therefore, the heat flux from the thermal bath to the particle equals the difference

$$\dot{q}(x; \sigma) = v\sigma U'(x) - (U'(x))^2 + T U''(x) - (-v\sigma U'(x) + v^2)$$
$$= 2v\sigma U'(x) - (U'(x))^2 + T U''(x) - v^2. \quad (4)$$

There is an alternative method to obtain that same expression which assumes that we can view the dynamics (1) as motion in the flashing potential

$$\Phi(\sigma, x) = -v\sigma x + U(x), \qquad \sigma = \pm 1, \quad (5)$$

with the dynamics (again taking $\gamma = 1$)

$$\dot{x}_t = -\Phi'(\sigma_t, x_t) + \sqrt{2T} \, \xi_t, \quad (6)$$

for a general flashing potential $\Phi$, while equivalent with (1) for RTPs in a confining potential $U$. Then, we simply find the heat from the change in energy at fixed $\sigma$,

$$\Phi(\sigma, x_t + dx_t) - \Phi(\sigma, x_t) = \left[ -(\Phi'(\sigma, x_t))^2 + T \Phi''(\sigma, x_t) \right] dt + \sqrt{2T} \, \Phi'(\sigma, x_t) \xi_t dt. \quad (7)$$

Upon substituting (5) in the drift part of (7), the expected heat flux to the particle is

$$\dot{q}(x; \sigma) = -(-v\sigma + U'(x))^2 + T U''(x) = -v^2 + 2v\sigma U'(x) + (U'(x))^2 + T U''(x), \quad (8)$$

exactly equal indeed to (4). Note that since we only need the mean values of heat (along transformations between nearby steady states), we do not have to introduce the heat and work as path-dependent quantities, as e.g. in stochastic thermodynamics [34, 35] where a Stratonovich integral is used for the heat. However, as it should be clear from the above derivation, we are entirely consistent on the level of statistical expectations.

Note that since we only need the mean values of heat (along transformations between nearby steady states), we do not have to introduce the heat and work as path-dependent quantities, as e.g. in stochastic thermodynamics [34, 35]. However, as it should be clear from the above derivation, we are entirely consistent on the level of statistical expectations.

### 2.3 Out-of-equilibrium calorimetry

We refer to [23, 36, 37] for the initial theory and basic examples of nonequilibrium heat capacities $C(T)$ as a function of bath temperature $T$. The idea is to estimate the quasistatic heat excess $\delta Q^{\text{ex}}$ after a small temperature change $\delta T$, while holding constant a given set of system and environment parameters:

$$C(T) = \frac{\delta Q^{\text{ex}}}{\mathrm{d}T}. \tag{9}$$

See also [38] for the general setup, including a motivation for the type of excess heat used in the present paper. In particular, we have the general result, Eq. III.2 in [38], that

$$\delta Q^{\text{ex}} = -\mathrm{d}T \left\langle \frac{\mathrm{d}V}{\mathrm{d}T} \right\rangle^s, \tag{10}$$

where the expectation $\langle \cdot \rangle^s$ is in the stationary distribution and $V$ is the quasi-potential defined from

$$V(\sigma, x) = \int_0^\infty \mathrm{d}t \left[ \langle \dot{q}(x_t; \sigma_t) - \dot{q}^s \mid \sigma_0 = \sigma, x_0 = x \rangle \right] \tag{11}$$

with stationary heat flux $\dot{q}^s = \langle \dot{q}(x; \sigma) \rangle^s$ and where the last expectation in (11) is for the process (1) or (6) starting from $(\sigma, x)$.

In equilibrium, the excess is just the heat produced in a reversible transformation, $\delta Q^{\text{ex}} = \delta Q = C(T)\mathrm{d}T$ and the quasi-potential $V(x) = U(x) - \langle U \rangle^s$ when the propulsion speed $v = 0$ (no dependence on $\sigma$). We emphasize that the nonequilibrium contribution cannot be reduced to a simple "thermodynamic" form; even in the quasistatic regime, excess heat over temperature does not need and typically will not be an exact differential [39, 40], except close-to-equilibrium [41–44]. In addition, for active matter, there is the additional difficulty that the position-process is not autonomous (i.e., not Markovian) while the theory in [23, 36, 37] was initially developed for Markov processes.

In the Appendix A we use the equations (9)–(10) to derive the heat capacity for an exactly solvable model. However, in the bulk of the paper, to measure or to compute (9) we apply AC-calorimetry [23, 45] where we vary the bath temperature at a given frequency $\omega \neq 0$, e.g. $T_t = T + \delta T \sin \omega t$. After the system relaxes (in a time that we assume is short compared with the ratio of excess heat to steady power dissipation), we measure the time-dependent heat flux $\dot{q}(t)$, in our case obtained by taking the steady expectation of (4). In linear order and neglecting $O(\omega^2, (\delta T)^2)$, we have the relation

$$\dot{q}(t) = \dot{q}^s + \delta T \left[ \Sigma_1(\omega, T) \sin(\omega t) + \Sigma_2(\omega, T) \cos(\omega t) \right], \tag{12}$$

defining $\Sigma_{1,2}(\omega, T)$ as the in- and out-phase components of the temperature-sensitivity of the dissipation. We assume that the temperature-heat admittance decays fast enough in time. The main difference with equilibrium calorimetry is that the DC-part $\dot{q}^s$ no longer vanishes. Around

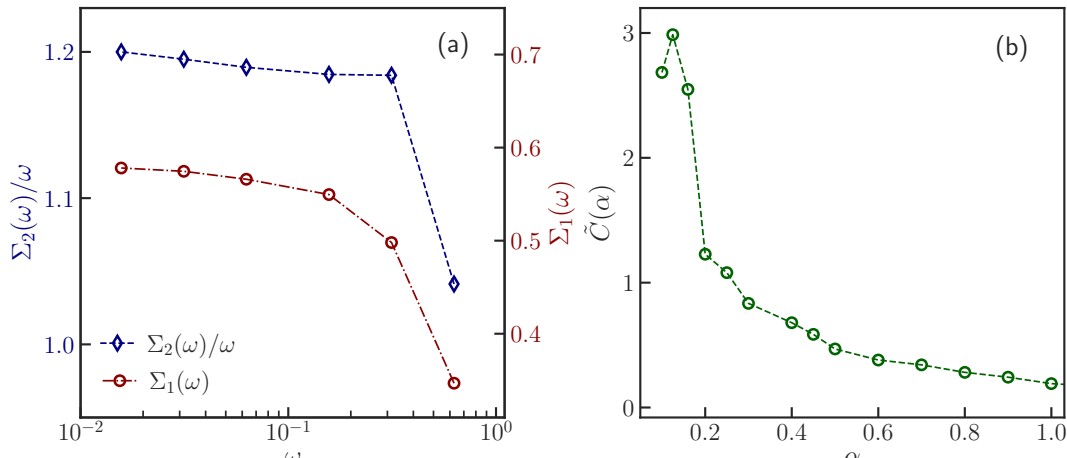

Figure 1: RTPs in a sinusoidal potential. (a) cf. (12)-(13): the in-phase ($\Sigma_1(\omega)$, as maroon circles) and out-of-phase ($\Sigma_2(\omega)$, as blue diamonds) amplitudes of the heat current at $T = 0.15, \alpha = 0.5, E_0 = 0.5$ and $v = 1$. (b) cf. formula (14): change in excess heat per change in tumbling rate $\alpha$ at $T = 0.1, v = 1.0$ and $E_0 = 0.5$. For both the plots system size is kept fixed at $L = 20$.

a steady nonequilibrium condition, the latter provides the dominant (for $\omega \to 0$) contribution to the heat flux, whereas the heat capacity (9) becomes the next correction. Indeed, the low-frequency asymptotics of the heat current (12) is, in the same linear order,

$$\dot{q}(t) = \dot{q}^s + \delta T \left[ B(T) \sin(\omega t) + C(T) \omega \cos(\omega t) \right], \tag{13}$$

or, $\Sigma_1(\omega, T) = B(T) + O(\omega), \Sigma_2(\omega, T) = C(T) \omega + O(\omega^2)$ with $C(T)$ as in (9) and where $B(T)\delta T = \dot{q}^s(T + \delta T) - \dot{q}^s(T)$ where we indicated the dependence on temperature in the stationary heat flux. The method explained in [23, 45] but, in essence, going back to the work of Sullivan & Seidel in [24] and to [25] for equilibrium processes, is applied in each of the nonequilibrium model systems we consider below.

In summary and looking back at (4), to compute the heat capacity through the steady heat flux (13), we thus need to evaluate

$$\dot{q}(t) = \left\langle 2v\sigma_t U'(x_t) - U'^2(x_t) + T_t U''(x_t) - v^2 \right\rangle_t,$$

for sufficiently large $t > 0$, where also the process average $\langle \cdot \rangle_t$ uses the slowly varying temperature $T_t = T + \delta T \sin \omega t$ to replace $T$ in the strength of the thermal noise in (1).

## 3  Run-and-tumble in a periodic potential

If moving on the circle we must have that $U$ is periodic in $x$. Consider then an RTP for position $x_t$ on the circle of length $L$, and moving in a sinusoidal potential. The dynamics is given by (1) with $U(x) = E_0 \sin(2\pi x/L)$. Note again that we can use (2) to understand the dependence on $L$, as it has to be compared with $v/\gamma\alpha$ for fixed $E_0$. Indeed, at fixed $E_0$, all dependence comes from the ratio between the persistence length $v/\alpha$ and $L$.

### 3.1  Simulation results

We fix $L = 20$ and we follow the scheme of Section 2.3. What is needed to simulate is the steady expectation of (4), and to fit it with (13).

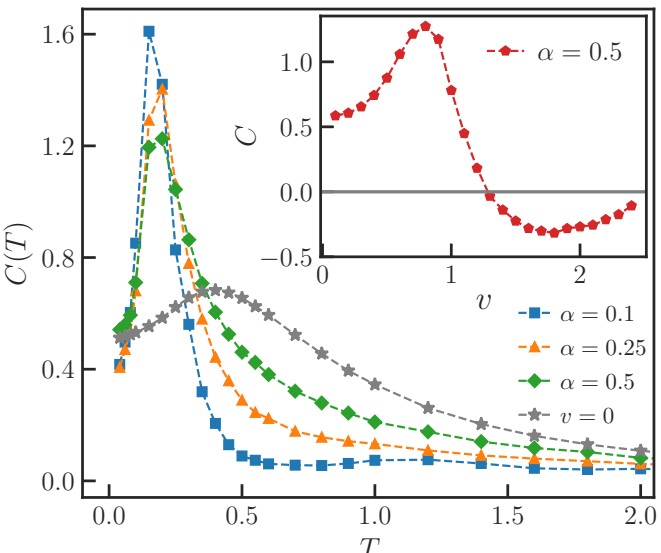

Figure 2: Heat capacity of RTPs for different tumbling rates, moving in a sinusoidal potential with amplitude $E_0 = 1.0$ and propulsion speed $v = 1.0$. The grey line corresponds to the $v = 0$ (equilibrium) case. The inset shows the variation of the heat capacity with $v$ for fixed $E_0 = 0.5$, tumbling rate $\alpha = 0.5$ and at temperature $T = 0.1$. System size is kept fixed at $L = 20$.

Fig. 1(a) shows the in- and out-of-phase components of the frequency-dependent heat current, defined in Eq. (12) for temperature $T = 0.3E_0$. The $\omega \downarrow 0$ limit of $\Sigma_2(\omega)/\omega$ gives the heat capacity, as follows from AC-calorimetry.

Fig. 2 shows the heat capacity for different tumbling rates. Interestingly, $C(T)$ has a sharp peak at around $T \simeq E_0/5$. The peak decreases and shifts towards higher temperature for increased tumbling rate $\alpha$. The peak value grows with the persistence to yield a significant magnification of the low-temperature heat capacity ($T < 0.3E_0$) with respect to equilibrium (= grey line in Fig. 2). Of equal interest, from Fig. 2 inset, the heat capacity gets negative when the propulsion speed $v$ is large enough to reach kinetic energies above the barrier height $E_0$.

When changing the tumbling rate $\alpha$ at constant ambient temperature, there is a change in excess heat as well. We can understand it as a nonequilibrium latent heat $\tilde{C}(\alpha)$, the change of the excess heat per tumbling rate. As in (13), we obtain it by applying a sinusoidal modulation $\alpha(t) = \alpha + \sin(\omega t)\delta\alpha$ for which the heat current becomes

$$\dot{q}(t) = \dot{q}^s + \left[ \tilde{B}(\alpha)\sin(\omega t) + \tilde{C}(\alpha)\,\omega\cos(\omega t) \right] \delta\alpha + \mathcal{O}(\omega^2). \tag{14}$$

Fig. 1(b) depicts $\tilde{C}(\alpha)$ for $T = E_0/5$, showing a sharp increase for smaller tumbling rates $\alpha$.

## 3.2  Discussion

The negativity of the heat capacity for larger propulsion speeds, as seen in the inset of Fig. 2 follows from the fact, that for larger speeds $v$, the particle gets a more uniform spatial distribution, as the barrier of size $E_0$ is easily overcome. In that sense, even though the ambient temperature may be low, the occupation statistics resembles (effectively for larger $v$) a profile as if in contact with a high-temperature bath. All the same, however, the heat released to the environment remains high. The higher $v$, the more heat is released to an even hotter

bath. That anticorrelation is at the origin of the negative heat capacity. We come back to it in Section 5.

In Fig. 2 we see that $C(T)$ has a pronounced Schottky-like anomaly [46], showing as a sharp peak at around $T \simeq E_0/5$. It indicates the presence of an energy scale $E_0$ for the height of hills separating discrete wells for possible low-temperature positions. At low temperatures, as the temperature increases, the particle is suddenly able to reach other valleys, which creates the peak. After all, $v$ has to be larger than $2\pi E_0/L$ to have a nonzero current at zero temperature. Because of the nonzero propulsion $v$, that transition comes earlier compared to equilibrium and happens over a shorter temperature-interval. The peak decreases and shifts towards higher temperature for increased tumbling rate $\alpha$ (lower persistence).

## 4 Run-and-tumble in a double-well potential

### 4.1 Model

Run-and-tumble particles in one-dimension and subject to an external confining potential have been studied for their static and dynamical properties; see e.g. [47, 48] and [49] for the non-Boltzmann stationary distribution at zero temperature. Heat capacities have never been computed, however. Here we consider RTPs confined by a symmetric double-well potential $U(x) = E_0(x^4/4 - x^2/2)$ in (1). The barrier height between the two wells is $\Delta = E_0/4$. The potential $U$ may arise as effective interaction in a mean-field description and may thus depend on the density profile as well, as relevant for, e.g., higher-dimensional motility-induced phase transitions [14].

Here we can view (1) as the motion (6) in a flashing potential

$$\Phi(\sigma, x) = E_0\left(\frac{x^4}{4} - \frac{x^2}{2}\right) - v\sigma x\,. \tag{15}$$

In other words, we have a flashing asymmetric (by $v$) double–well potential and we can also apply (8).

To compute the heat capacity through (13), we thus need to evaluate, via simulation, the steady heat flux

$$\dot{q}(t) = \left\langle -\Phi'^2(\sigma_t, x_t) + T_t\,\Phi''(\sigma_t, x_t)\right\rangle_t\,. \tag{16}$$

### 4.2 Simulation results

The resulting heat capacity figures in Fig. 3 (a) for different propulsion speeds $v$. Fig. 4 gives that heat capacity for a broad range of tumbling rates $\alpha$, showing also the broader peak at a temperature $T \simeq E_0/2$ for $\alpha \leq 0.5$. In Fig. 3(a), we see the appearance of negative heat capacities when $v^2/\alpha$ (see (2)) gets large compared to the barrier height $\Delta$.

### 4.3 Discussion

Fig. 5 shows a comparison between equilibrium and highly persistent RTPs. The upper curve (blue circle) is the heat capacity of a (fixed) asymmetric double–well potential (aDW),

$$\Phi(-1, x) = E_0\left(\frac{x^4}{4} - \frac{x^2}{2}\right) + vx\,, \tag{17}$$

which is (15) for fixed $\sigma_t \equiv -1$. Compare with Fig. 4 for the small $\alpha$−regime. For the two upper curves, the propulsion speed $v$ is large so that $\Phi(\pm, x)$ has a single minimum at a certain $\pm x^*(v)$. For low temperatures and small tumbling rate $\alpha$, the dynamics is quickly relaxing

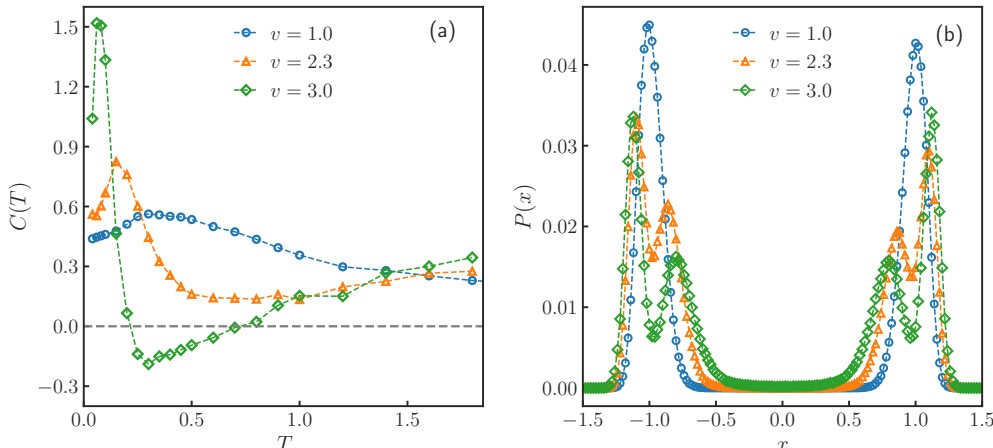

Figure 3: (a) Heat capacity for RTPs in a double-well potential with barrier height $\Delta = 2.5$ for different propulsion speeds $v$ at tumbling rate $\alpha = 0.5$. (b) Corresponding steady state occupation density $P(x)$ for different propulsion speeds. Here $E_0 = 10.0$, and $T = 0.1$.

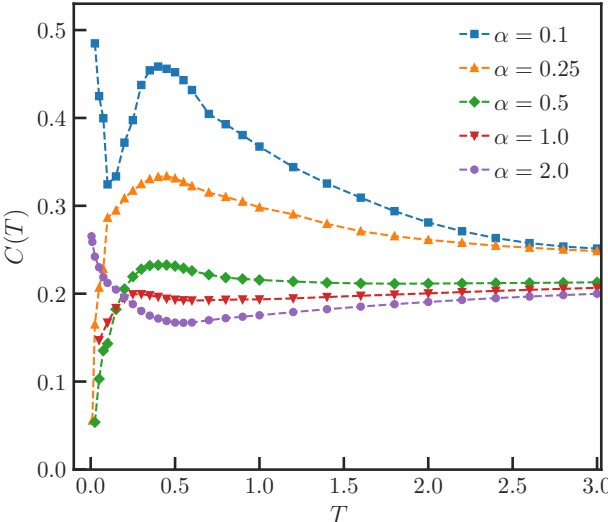

Figure 4: Heat capacity of RTPs confined by a double-well potential plotted for different tumbling rates at $v = 1.0$ and barrier height $\Delta = 0.25$, $E_0 = 1.0$.

to the neighborhood of the potential minimum $\pm x^*$ and we find that the heat capacity of the RTPs resembles the DW-curve representing Gaussian fluctuations in an effective quadratic potential, $C \sim 1/2$. When temperature increases, the heat capacity qualitatively picks up the behavior of the aDW-curve as the passive fluctuations start to be governed by the subquadratic segment in (15). At high-T, we get fluctuations in an effective quartic potential, $C \sim 1/4$. The lower curve (green squares) has $v = 0$ (DW); see also [50].

Interestingly, the heat capacity detects the zero-temperature shape transition, [47, 48, 51], from the behavior of the heat capacity at low temperature: Fig. 3(b) shows the stationary distribution and how it changes at $T = 0.1$ for the same parameters as in Fig. 3(a). When $v$

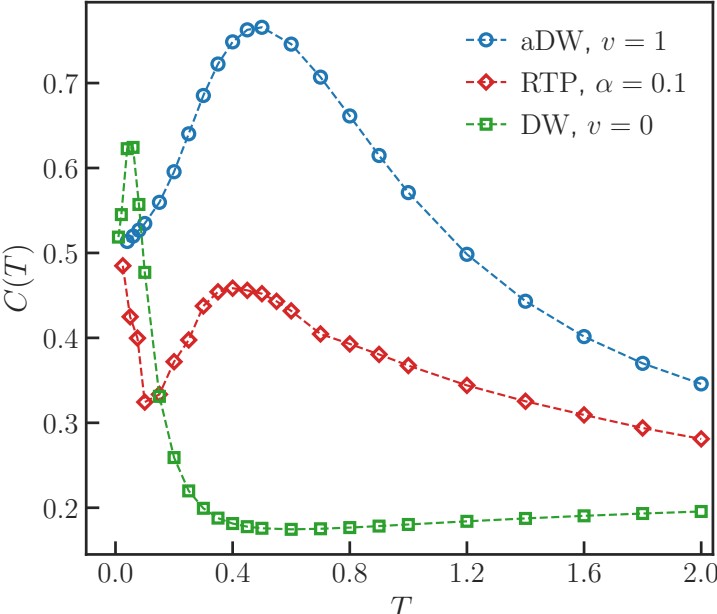

Figure 5: Heat capacities for particles in an asymmetric double well (upper curve), for RTPs in a double-well potential at $v = 1.0$, barrier height $\Delta = 0.25$, $E_0 = 1.0$ and $\alpha = 0.1$, and for particles in a symmetric double well (lower curve). See around (15).

is still small, the occupation is bimodal at low temperatures (corresponding to the two wells, as in equilibrium). As we increase $v$ (going active), there appears a bimodality in each well, leading to four local maxima in the occupation distribution. Those "edge states" combine with a sharper and higher low-temperature peak in the heat capacity.

To shed more light on the influence of the nonlinearity, we consider an exactly solvable case with

$$\Phi(\sigma, x) = \frac{k}{2}(1 + \varepsilon\sigma)x^2, \qquad 0 \leq \varepsilon \leq 1, \tag{18}$$

where, as before, $\sigma = \pm 1$ is flipping randomly at rate $\alpha$. This model does not represent RTPs, but it has a flashing potential with dynamics (6) and therefore is relevant for comparison and inspiration. We refer to Appendix A for the calculation.

The stationary energy at fixed temperature $T$ is $E^s = T/2$. It depends only on temperature, giving the (false) impression that the system partitions energy in the same way as in equilibrium. Yet the heat capacity becomes nontrivial, though still temperature–independent, and it reveals a fundamentally different thermal response than for a passive particle,

$$C(T) = \frac{1}{2}\left[1 + \frac{\varepsilon^2(1 + 2z)}{(1 + z(1 - \varepsilon^2))^2}\right], \tag{19}$$

where the (dimensionless) persistence factor $z = k/\alpha$ appears; see (A.8). We thus distinguish the following regimes:

a. $\varepsilon = 0$ (vanishing switching amplitude) is the equilibrium case of an overdamped diffusing particle confined by a harmonic potential. In this case the stationary heat flux $\dot{q}^s = 0$ and the heat capacity $C = \frac{1}{2}$ as expected from equipartition.

b. $\varepsilon = 1$ corresponds to one-dimensional overdamped diffusion in a quadratic potential which is randomly switched on and off. Unlike equilibrium there is a nonzero stationary heat flux $\dot{q}^s = -kT$ and $C = 1 + z$. The heat capacity depends on the persistence factor $z$ which is inversely proportional to the switching rate of the potential.

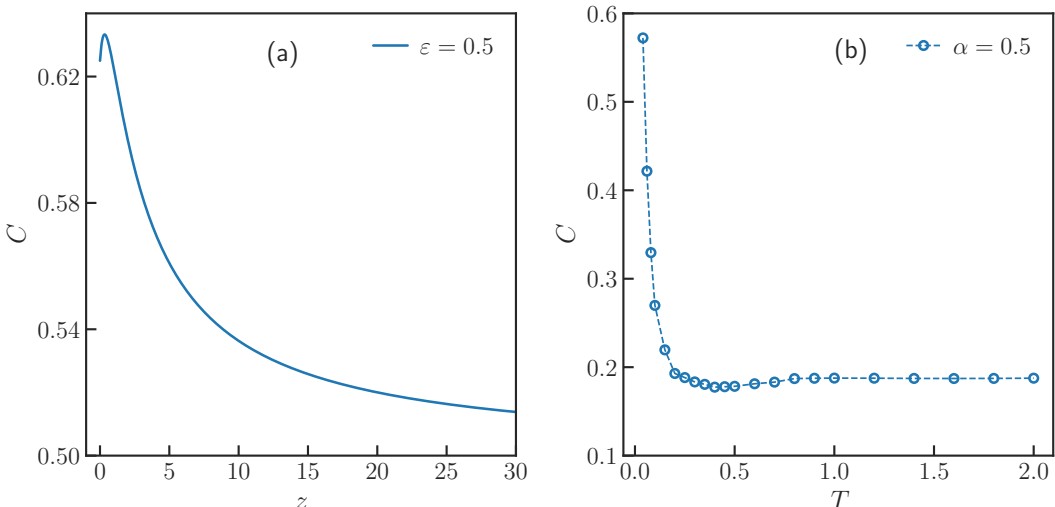

Figure 6: (a) Heat capacity (19) of a particle moving in a flashing harmonic potential (18), plotted as a function of persistence factor $z = k/\alpha$ for $\varepsilon = 0.5$. For $z \to 0$, $C \to 5/8$. (b) Heat capacity of a particle moving in a flashing symmetric double-well potential (20), plotted as a function of temperature for flashing rate $\alpha = 0.5$, $k = 1.0$.

c. $z \to 0$ implies a very fast switching of the potential, i.e., vanishing persistence. Interestingly, there is a nonzero heat flux $\dot{q}^s \to -kT\,\varepsilon^2$ (i.e., the system truly remains away from equilibrium), and with a heat capacity $C \to \frac{1}{2}(1 + \varepsilon^2)$ that is always larger than the equilibrium one ($\varepsilon = 0$), indicating an enhanced thermal response to temperature variations. Special attention is paid to that case in the Appendix A.

d. $\varepsilon < 1$, $z \to \infty$ is the case of infinite persistence. In this case, the system can effectively be treated as in equilibrium for a harmonic potential: the stationary heat flux $\dot{q}^s \to 0$ and $C \to \frac{1}{2}$.

We repeat that the equilibrium case is clearly separated from the flashing case, despite the "equipartition" $E^s = T/2$. In Fig. 6(a) we plot the heat capacity as a function of $z$ for $\varepsilon = 1/2$. In particular, we see that we can obtain the persistence factor from the heat capacity.

To contrast, Fig. 6(b) gives the heat capacity for a particle in a flashing symmetric double well,

$$\Phi(\sigma, x) = \frac{k}{2}\left(1 + \frac{\sigma}{2}\right)\left(\frac{x^4}{4} - \frac{x^2}{2}\right), \qquad \sigma = \pm 1. \tag{20}$$

That double-well case cannot be solved exactly and the AC-numerical work is able to show how the (nonequilibrium) heat capacity nontrivially depends on (low) temperature.

## 5 General features

### 5.1 Negative heat capacity

In the equilibrium canonical ensemble, the heat capacity is proportional to the variance in energy and is therefore always positive. As we see from (10), in nonequilibrium, the heat capacity is a covariance between the quasi-potential $V$ and the change in temperature of the

stationary distribution $\rho$; see also Eq. III.5 in [38],

$$C(T) = \left\langle V \; ; \; \frac{\mathrm{d}}{\mathrm{d}T} \log \rho \right\rangle .$$

That need not be positive. It involves the correlation between heat and occupation statistics, and those can be negatively correlated. Indeed, in stationary nonequilibrium, Clausius heat-related entropy, as to be discussed in Section 5.3, and Boltzmann entropy no longer coincide. The anticorrelation happens when there is a population inversion so that higher temperature brings particles in lower energy states, meaning that the heat flow to the reservoir increases even though its temperature got higher. Flashing or active forces are especially effective for that scenario when the driving amplitude (such as the propulsion speed) gets large with respect to the energy barriers.

We see the phenomenon of negative heat capacity in Fig. 3 and in the inset of Fig. 2. In all cases, the heat capacity gets negative when the propulsion speed $v$ is large enough to reach kinetic energies above a barrier height.

## 5.2 Comparing with equilibrium

We saw in Section 4.3 the exactly solvable case of a flashing potential for which the stationary energy is $E^s = T/2$. As it only depends on temperature, it reminds us of the equipartition relation, especially as we deal there with a harmonic potential. Nevertheless, the heat capacity (19) is nontrivial, and fundamentally different from equilibrium.

The effect of persistence is visible in the heat capacity, as shown in Fig. 6(a). In that way, we can diagnose the state of the active system. For example, in Fig. 2 the peak value grows with the persistence to yield a significant magnification of the low-temperature heat capacity ($T < 0.3E_0$) with respect to equilibrium (= grey line in Fig. 2).

## 5.3 Heat-related entropy

Entropy originates in the Clausius heat theorem, gets a statistical meaning as a measure of phase volume, and gives rise to statistical forces. Such a protean entropy does not exist for genuine nonequilibria, [39,40]. Yet, a heat-related entropy $\not\!\Delta S$ can be defined, also for active systems, by taking $\not\!\Delta S$ as operationally defined from the heat capacity, as we would do for an equilibrium *entropy*:

$$\not\!\Delta S(T) = \int_{T_0}^{T} \mathrm{d}T' \, \frac{C(T')}{T'} , \tag{21}$$

for some reference (initial) temperature $T_0$. Note that (21) does not define entropy as a state function (and that motivates our notation with $\not\!\Delta$).

We plot $\not\!\Delta S(T)$ in Fig. 7 for RTPs in two different landscapes and for two tumbling rates. We only used the data for $C(T)$ (in Fig. 2 and Fig. 4) for temperatures $T > T_0 = 0.04$ and $0.025$ respectively. Note that in equilibrium, we always have that $\int_0^\infty \mathrm{d}T \, \frac{C(T)}{T}$ equals the infinite-temperature entropy. For active systems, however, that integral depends on the persistence $\alpha^{-1}$.

## 6 Conclusion

(Thermal) active systems are in physical contact with (at least) two reservoirs: one which is often chemical or radiative and a source of low entropy, and one which can be identified with a thermal bath or environment in which energy gets dissipated. Perturbing the temperature,

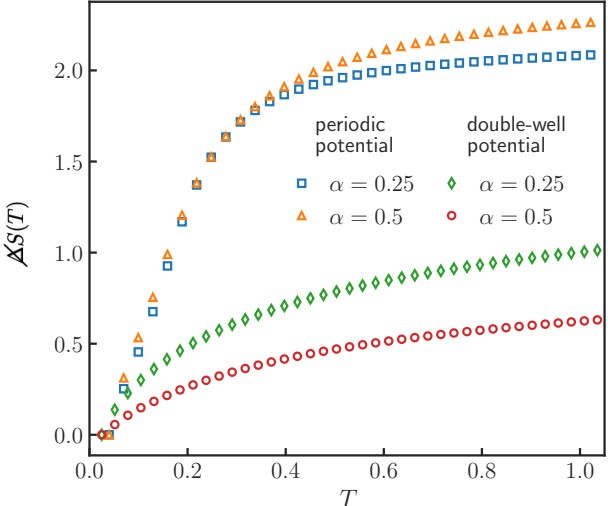

Figure 7: Heat-related entropy (21) as a function of temperature for RTPs moving in a sinusoidal potential (upper curves) (parameters: $v = 1$, $E_0 = 1$, $L = 20$) , and confined by a double-well potential (lower curves) (parameters: $v = 1$, $E_0 = 1$, $\Delta = 0.25$). We emphasize that (21) refers to a change in temperature only.

the heat capacity measures the excess heat in addition to the steady ever-existing dissipation. This paper has indicated how it may depend on activity parameters.

For the first time for active systems, we have observed numerically Schottky-like anomalies and a regime of negative heat capacity where an increased environment temperature enhances the excess dissipation. We are confident that low-temperature active materials show thermal characteristics as in the discussed model systems, to become a new and fascinating subject of investigation in materials science. Also for bio-systems, in a more restricted range of temperatures, the results indicate how heat capacity can serve as a diagnostic tool for activity.

We note that calorimetry, even earlier than spectroscopy, has been a major tool in disclosing the relevant degrees of freedom of a material and how those can be excited depending on temperature. In our model systems, we have focused exclusively on translational degrees of freedom (locomotion in one dimension) which is a great simplification. Obviously, other *internal* degrees of freedom will contribute as well to the heat capacity, just as in equilibrium. It is therefore not entirely clear whether calorimetric measurements will have enough precision to distinguish the equilibrium from the nonequilibrium effects. Still, as shown in the model systems, giant low-temperature magnifications and negativity of the heat capacity can be expected to stand as clear signs and diagnostic tools for the activity.

Biophysical experiments include [6,7] but using AC-calorimetry as numerically pioneered here for active systems, would be very welcome to carry that program, to verify those predictions and hence to continue the old adage that *Even fire is ruled by numbers*[1] in the physics of active and living materials as well.

---

[1]Et ignem regunt numeri, citation attributed to Plato on the title page of Ref. [52].

# A  Flashing potential

We derive the results of Section 4.3, where we consider a flashing harmonic potential for an evolution described by (6),

$$\dot{x}_t = -\partial_x \Phi(\sigma_t, x_t) + \sqrt{2T}\,\xi_t, \qquad \Phi(\sigma, x) = \frac{k}{2}(1 + \varepsilon\sigma)\,x^2. \tag{A.1}$$

The parameter $\varepsilon \in [0, 1]$ prescribes the ratio by which the potential is turned off.

We employ here the method outlined in (9)–(11). We use the expected heat flux defined from (7),

$$\dot{q}(x; \sigma) = -(\Phi'(\sigma, x_t))^2 + T\,\Phi''(\sigma, x_t) = -k^2(1 + \varepsilon^2)x^2 - 2\varepsilon k^2\sigma x^2 + kT(1 + \varepsilon\sigma).$$

Under the stationary distribution,

$$\langle x^2\rangle^s = \frac{1+z}{1+z(1-\varepsilon^2)}\frac{T}{k}, \tag{A.2}$$

$$\langle \sigma x^2\rangle^s = -\frac{z\varepsilon}{1+z(1-\varepsilon^2)}\frac{T}{k}, \tag{A.3}$$

where $z = k/\alpha$ is the dimensionless persistence factor. Note that for flashing rate $\alpha \downarrow 0$, the variance (A.2) is the sum of the variances corresponding to $\sigma = \pm 1$, while the correlation (A.3) remains different from zero for $\varepsilon \neq 0$. On the other hand, the stationary energy is

$$E^s = \frac{k}{2}\big(\langle x^2\rangle^s + \langle \varepsilon\sigma x^2\rangle^s\big) = \frac{T}{2} \tag{A.4}$$

and depends only on temperature.

The stationary heat flux to the system equals

$$\dot{q}^s = -kT\,\frac{\varepsilon^2}{1+z(1-\varepsilon^2)}. \tag{A.5}$$

The quasi-potential of (11), [36, 37], can be found as the solution to the equation $(LV)(\sigma, x) = \dot{q}^s - \dot{q}(x; \sigma)$ (unique on the appropriate functional space, $\|Y\|^2 = \langle Y^2\rangle < \infty$), where $L$ is the backward generator corresponding to (A.1). Substituting the *Ansatz*

$$V = \tilde{V} - \langle\tilde{V}\rangle, \qquad \tilde{V}(\sigma, x) = c\,x^2 + d\,\sigma x^2 + g\,\sigma,$$

we find the corresponding coefficients

$$c = \frac{k}{2}\frac{1+\varepsilon^2+z(1-\varepsilon^2)}{1+z(1-\varepsilon^2)}, \quad d = \frac{k}{2}\frac{z\varepsilon(1-\varepsilon^2)}{1+z(1-\varepsilon^2)}, \quad g = -\frac{T}{2}\frac{z\varepsilon}{1+z(1-\varepsilon^2)}. \tag{A.6}$$

Furthermore,

$$\langle\tilde{V}\rangle^s = \frac{T}{2}\left[1 + \frac{\varepsilon^2(1+2z)}{(1+z(1-\varepsilon^2))^2}\right], \tag{A.7}$$

so that the quasi-potential $V(\sigma, x)$ remains different from $\Phi(\sigma, x) - E^s$ even for $\alpha \downarrow 0$, as long as $\varepsilon \neq 0$:

$$\lim_{\alpha \downarrow 0} V(\sigma, x) = \Phi(\sigma, x) - \frac{T}{2}\left(1 + \frac{\varepsilon\sigma}{1-\varepsilon^2}\right).$$

Finally, the steady heat capacity (9) is independent of temperature and equals

$$\begin{aligned}
C &= -\left\langle\frac{\partial V}{\partial T}\right\rangle^s = \frac{\partial\langle\tilde{V}\rangle^s}{\partial T} - \left\langle\frac{\partial\tilde{V}}{\partial T}\right\rangle^s \\
&= \frac{1}{2}\left[1 + \frac{\varepsilon^2(1+2z)}{(1+z(1-\varepsilon^2))^2}\right].
\end{aligned} \tag{A.8}$$

Note that the last term on the first line is zero, since $\partial\tilde{V}/\partial T \propto \sigma$ and $\langle\sigma\rangle^s = 0$.

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
