# Peer review of "Calorimetry for active systems"

_SciPost Physics, doi:SciPost Phys. 14, 126 (2023)_

## Round 3 · Referee Report · Anonymous (Referee 1) · 2023-1-20

Report

The revisions to this manuscript have improved it significantly. There is a much more comprehensive discussion of the results, and their significance is much clearer. I have some small suggestions for minor revisions, see below.

My judgement is that the SciPost general criteria are now met. However, it is still not clear to me that the more subjective "expectations" for SciPost Physics are satisfied ("groundbreaking discovery" / "breakthrough" / etc).

I recommend publication, based on the general scientific criteria of clarity, accuracy, etc. I leave it to the editors to decide whether their expectations are met.

Requested changes

  1. For the numerical models, it would be very useful to identify the sets of dimensionless parameters that determine the model behaviour. The authors set the unit of time by setting gamma=1, but they are still free to set units of length and energy without affecting the physical behaviour. For example, it seems that the results of Sec III should depend on E0 and T only through their ratio E0/T (up to linear scaling of the heat capacity, etc). There is a similar question about dependence on system size: a relevant dimensionless parameter seems to be v/(L.alpha), the ratio of persistence length to system size. Does all dependence on L come through this parameter? Similar questions are relevant in Sec IV.

  2. Related to point 1, I suggest that /all/ model parameters be stated in the captions of all the figures. (Alternatively, reduce the equations to dimensionless form, so that it becomes sufficient to quote only the relevant dimensionless combinations.)

  3. equation (10) seems to have a mismatched [.

  • validity: high
  • significance: good
  • originality: high
  • clarity: high
  • formatting: good
  • grammar: good

Author:  Pritha Dolai  on 2023-01-30  [id 3283]

(in reply to Report 1 on 2023-01-20)

Referee's comment: " 1. For the numerical models, it would be very useful to identify the sets of dimensionless parameters that determine the model behaviour. The authors set the unit of time by setting gamma=1, but they are still free to set units of length and energy without affecting the physical behaviour. For example, it seems that the results of Sec III should depend on E0 and T only through their ratio E0/T (up to linear scaling of the heat capacity, etc). There is a similar question about dependence on system size: a relevant dimensionless parameter seems to be v/(L.alpha), the ratio of persistence length to system size. Does all dependence on L come through this parameter? Similar questions are relevant in Sec IV. "

Reply: As for the 'dimensions' and 'dimensionless dependencies' we have added Eq 2 in the manuscript. From there the dependencies are clear. We also add there what we do, referring to varying the tumble rate and the propulsion speed. We think it is also a matter of taste and analysis what setup one prefers. At any rate, now it is clear how to move in the parameter space. In that sense we also added a comment in the beginning of Section III to discuss the length-scale. Indeed, all dependence comes from the ratio between the persistence length $v/\alpha$ and $L$. Similarly, in Section IV. the barrier height must be compared with $v^2/\alpha$; see end of page 11 for an extra comment.

Referee's comment: "2. Related to point 1, I suggest that /all/ model parameters be stated in the captions of all the figures. (Alternatively, reduce the equations to dimensionless form, so that it becomes sufficient to quote only the relevant dimensionless combinations.) "

Reply: We have stated all model parameters in the captions of all the figures.

Referee's comment: " 3. equation (10) seems to have a mismatched [. "

Reply: We have corrected the typo in (now) Eq 11.

---

## Round 3 · Referee Report · Anonymous (Referee 2) · 2023-1-23

Report

The authors satisfactorily answered the referees' comments. I recommend that the paper is accepted for publication in the SciPost.

Optional suggestions (that can be addressed in the proofs stage): 1) I would encourage the authors to note that most of the stochastic thermodynamics literature uses the Stratonovich interpretation when writing the expression for the heat (this difference caused my initial confusion, for which I apologize). 2) It is not clear what does the statement "with the barrier also providing a discrete character in the possible positions" mean since the position is obviously a continuous variable.
  • validity: good
  • significance: high
  • originality: high
  • clarity: good
  • formatting: -
  • grammar: -

Author:  Pritha Dolai  on 2023-01-30  [id 3282]

(in reply to Report 2 on 2023-01-23)

We thank the referee for recommending our paper for publication.

Referee's comment: " 1) I would encourage the authors to note that most of the stochastic thermodynamics literature uses the Stratonovich interpretation when writing the expression for the heat (this difference caused my initial confusion, for which I apologize)."

Reply: We have added this note in the heat and work section (section IIB).

Referee's comment: "2) It is not clear what does the statement "with the barrier also providing a discrete character in the possible positions" mean since the position is obviously a continuous variable. "

Reply: We have rewritten the sentence making it more clear now.

---

## Round 3 · Author Response

Dear Editor,

We thank both the referees for their reviews. The comments of the referees suggest some clarifications, which we have provided now. We have made a major revision of the manuscript clarifying the points addressed by the referees. In fact, we have benefited much from the questions and remarks. The detailed replies to all the points raised by the referees are provided below.

We are pleased to submit the revised version which we believe is in a much better shape than the previous version.

Regards, Pritha Dolai, Christian Maes, Karel Netočný

Reply to Report of Referee 1 --

Referee's comment : This is an interesting manuscript about the exchange of heat between a non-equilibrium system and its environment: this is what is meant by calorimetry. Specifically, the authors analyse a quantity that they call the non-equilibrium AC heat capacity.

Reply: We are pleased that the referee finds our manuscript of interest. That is always encouraging.

Referee's comment : 1. The general model to be considered is eq(4). This includes as special cases run-and-tumble particles and (so-called) flashing potentials. The paper needs a clearer motivation for what these systems represent, and why they are interesting in this context. A brief review of some relevant literature would be appropriate, either in the introduction, or when the model is introduced.

Reply: Indeed, we have the paragraph (in the Introduction, starting with “The present paper takes... ) with some references [6, 8, 18–24]. We have added more specific motivation for considering RTPs in the beginning of Section II and Section IIA), stating that these models are paradigmatic and simple examples of self-propelled locomotion, simplifying features of some natural or artificial biophysical entities (bacteria, motors etc). While their mechanical properties are by now rather well understood, their thermal properties have not been analyzed in sufficient detail yet. We attempt to fill this gap by asking a natural calorimetric question, namely how these systems thermally (in terms of the heat current) respond to slow variations of ambient temperature. Still a broader motivation of our research is to build a collection of paradigmatic nonequilibrium models with well understood properties, which allow us to, e.g., learn how to detect different types of driving mechanisms of more realistic systems based on their thermal (and mechanical) response to external disturbances, which is precisely what is the calorimetry generally about.

Referee's comment : Related : the introduction seems to imply that systems with flashing potentials are "active systems". As far as I can see, a system with a flashing potential would be more naturally described as a system in a time-dependent external potential. (The term "active" is usually reserved for systems where particles inject energy locally, for example by self-propulsion.)

Reply: Correct, and we make clearer distinction between ‘flashing’ and ‘active.’ The paper is about ‘active’ calorimetry, but it turns out, as we now fully explain in Sections IIB and IVA, that there is a mathematical equivalence between the evolution of an active particle in a confining potential, and the evolution in a flashing potential. Specifically (as used in Section IVA), the switching of a flashing potential can be thought of as a direction-flipping event of one-dimensional run-and tumble particles. Of course, it remains true that the term “active” is generally used for self-propelled particles, and we avoid confusing the reader between flashing potentials and active systems. Nevertheless, we do present the exactly solvable model of Eq 17 and Eq A1 (flashing harmonic, which is not an RTP), because it fits naturally in the discussion of RTPs in a double well potential.

Referee's comment : Also, I believe that most studies of run and tumble particles are conducted without any thermal noise in the equation of motion (presumably because thermal diffusion is assumed to be negligible with respect to self-propulsion). The authors should clarify that their run-and-tumble differs from the majority of studies on such systems.

Reply: It is true that many studies of RTP are carried out at zero thermal noise, because they are mostly focused on mechanical properties of those active systems such as bacteria. We add noise (very little at low temperatures) since we are interested in thermal properties here. Even though thermal diffusion is mostly negligible for bacterial motion, they do exchange energy with the heat bath. Yet, for bacteria, the contribution to heat from translational motion is expected rather small and its detection remains a challenge – we are indeed aware of this problem. More generally (and more relevant), RTPs are paradigmatic for active components, also artificial ones all the way to nanoscales, and the influence of noise (e.g. thermal) is a very natural subject of study. We mention that more clearly now at the end of Section IIA, and in the introduction to Section II. We have added some references as well.

Referee's comment : 2. In general, there is very little discussion of the results. This needs to be expanded. For example: 2a. below eq (7), four cases are enumerated and named, but there is no discussion of the physical features of these cases. What are interesting/important properties of these cases and how do these manifest in C(T)?

Reply : We have added discussion sections IIIB, IVC and V. That way, issues are separated depending on the specific model and on what in general. Concerning the solvable example for the flashing potential of (now) Eq 17 and Eq A1, we added discussions on the various cases. In particular, starting after Eq 18 we give a detailed survey of various regimes ,calculated in detail in the Appendix, depending on the switching amplitude 'epsilon' and the dimensionless persistence factor 'z'. It demonstrates the richness of different patterns in thermal response and helps to build some intuition towards its interpretation, which we then also use for the results obtained by simulations on the other more realistic models.

Referee's comment : 2b. The introduction seems to imply that experimental measurements of C(T) would provide some interesting insight into properties of active systems. However, the authors do not discuss what insights (in any) are available from the numerical results of Sec III.B, for run-and-tumble particles. The same comment applies to Sec III.C. The authors need to add (at least) a paragraph to each of these sections, to explain what the reader is supposed to learn from the results.

Reply: Now we have added a detailed discussion after each of these numerical results for run-and-tumble particles moving in a periodic potential and a double well potential. We are aware that a detailed understanding of relation between the structure of a driven or active system and its thermal response is still not sufficiently understood (in contrast to equilibrium systems) and remains an interesting yet nontrivial open question. So far we are mostly learning via examples and that is why we believe that our numerical results, are the first data on active calorimetry, and are interesting even per se.
The main thing to learn is the relation between activity and the heat capacity, how it differs from equilibrium and how it depends on temperature. For example, there is the giant increase of heat capacity at low temperature (qualitatively preserving the Schottky anomaly) and the occurrence of negative heat capacity signaling a negative correlation between the excess heat and the occupation statistics. We also like that the presence of ‘edge states’ as previously observed for RTPs at zero temperature, also shows in their low-T heat capacity (Fig.3b).

Referee's comment : 2c. In some cases, negative values are found for the specific heat. This should presumably have some physical interpretation. (The oscillation in the heat current is out of phase with the temperature oscillation, even at low frequencies?) What are the physical mechanisms for this effect? Are the authors able to say anything generic about what kinds of system would exhibit negative non-equilibrium heat capacities? The authors might want to defer some of these issues to future work. However, my opinion is that the manuscript should not be accepted until the discussion of these points has been significantly improved.

Reply: We have added a subsection (Section VA) to explain negative heat capacity.

Referee's comment : 3. The possibility of experimental measurements is mentioned several times in the Introduction and conclusion. I am not sure how this would work in practice. Any experimental system would necessarily have many more degrees of freedom than the one-dimensional position co-ordinate considered here. I think the authors should briefly discuss how their results would change if the particle that they consider has some internal degrees of freedom. Even if the equations of motion of these internal co-ordinates are decoupled from eq(4), they can still exchange energy with the heat bath. Do the authors think that calorimetric measurements could be performed with enough precision to separate the non-equilibrium heat transfer from the equilibrium-like (but still frequency-dependent) exchange of heat between the internal degrees of freedom and the heat bath?

Reply: That is indeed a challenging and highly relevant and exciting discussion. We all agree it would be interesting to start the experiments (and contacts have been made), and we must wait and see. It will be interesting to see deviations from the results in the paper, as they will mean something, but, on the other hand, we believe various of the main features (giant magnification of Schottky peak and negative heat capacities) will remain. Calorimetry is exactly scanning what static and dynamical degrees of freedom become available, as the external temperature varies. We think the present paper is exactly showing the way (the AC calorimetric way) of doing the experiments. The paper is therefore to be seen as pioneering exploration, with proof of principle (on the level of simulations) of interesting things to be expected. We add more on these issues in the Conclusion now.

Referee's comment : Smaller points: 4. Is eq (2) supposed to be valid up to corrections at O(delta T)^2 ? It is stated explicitly for eq (3) that higher order terms have been neglected, why is there no similar statement for (2)?

Reply: It is correct that (former) Eq 2 ( = Eq 11 in the present version) has corrections at O(delta T)^2. We have added it in the text now.

Referee's comment : 5. Comparing eqs (2) and (3), are we supposed to infer some relationship like sigma1(omega) = B + O(omega^2) sigma2(omega) = C omega + O(omega^3) If this is the case, perhaps it can be stated. (More generally, the relationship between these two equations should be explained more clearly.) Also, if B and C are functions of T (as this implies), perhaps it is better to write sigma1 and sigma2 as functions of (omega,T)?

Reply: Now we have written Sigma1 and Sigma2 in terms of B and C. Sigma1(omega, T) =B(T) +O(omega) Sigma2(omega, T) = C(T) omega + O(omega^2)

Referee's comment : 6. In equation (10), is the v that appears in this potential the same as the the velocity 'v' that appears in equation (4)? Also, this U is a function of x alone, but the text just below seems to refer to U(x,eta) with eta=\pm1. Perhaps the authors could just write the full equation of motion for this system, that would avoid any confusion.

Reply: Indeed, that was confusing. We have made it clear now, in particular by not mixing up ‘active’ and ‘flashing’.

Reply to Report of Referee 2 --

Referee's comment : The paper introduces notions of heat and work that seem different from those used in standard stochastic thermodynamics and in its application to active matter systems. The authors do not sufficiently motivate their choice and do not compare/contrast it with that of stochastic thermodynamics. Classic papers by Sekimoto and the more recent EPL Perspective by Fodor and Cates are not cited. While it is possible that the authors' definitions are superior, the difference with earlier works should be discussed. 1) The definitions of work and heat should be motivated and compared with those used in earlier literature.

Reply: The definition of work and heat are discussed in detail in (the new) Section IIB and compared with the existing literature and the corresponding references have been added. In particular, we only need the instantaneously expected heat fluxes, which is not the one used in stochastic thermodynamics, however we are entirely consistent (on the level of mean values) with the standard energetic picture of stochastic thermodynamic as introduced by Sekimoto. See also the end of Section IIB.

Referee's comment : 2)The physical meaning of the negative heat capacity should be discussed.

Reply: We have added a discussion about the origin of negative heat capacity in Section VA.

Referee's comment : 3) The paper should be carefully re-read and potentially confusing colloquial statements should be re-written.

Reply: We have rewritten the paper and removed the confusing statements.

Referee's comment : Minor comments: 1) There are some potentially confusing phrases in the paper. For example the meaning of "maintaining relatively large heat" is unclear since we teach from introductory thermodynamics that heat is a mode of energy transfer rather than a state function. On the same page there is "heat capacity will differ between live matter and its unorganized mixture of molecules" which suggests that live matter is always organized and passive (dead) matter always isn't.

Reply: We have rephrased the sentences (there and elsewhere).

Referee's comment : 2) What is the meaning of "quasi-entropy"?

Reply: The section VC on entropy is motivated by many questions, how we can do calorimetry without entropy… We emphasize that the protean concept of entropy no longer holds in nonequilibrium. Heat fluxes are not just governed by the occupation statistics, of course, and its information-theoretic entropy does not play the role of a meaningful thermodynamic potential when out of equilibrium. We no longer use the word ‘quasi-entropy’ but have chosen now for ‘heat-related entropy’ defined in Eq 20. As we write, that heat-related entropy uses formally the same formula for computing the entropy from heat capacity, as one would do in equilibrium. It is indeed not essential at all for the paper, but from experience, many people ask for it.

---

## Round 3 · List of Changes

Here we summarize the main changes:

  1. we have added a separate section and detailed discussion (Sction IIB) on the heat and work that are involved in the calorimetric calculations.
  2. we spent a separate paragraph (Section VA) on discussing the occurrence of negative heat capacity.
  3. we have added separate sections on the discussions (Sections IIIB, IVC and V) of the phenomena, both for the specific cases and in general.
  4. we have elaborated on the use of run-and-tumble particle models, their naturalness and relevance for the purpose.
  5. we have added some relevant references.

---

## Editorial Decision

published